# Assessment of the Water Footprint in Low-Income Urban Neighborhoods from Developing Countries: Case Study Fátima (Gamarra, Colombia)

Brayan Alexis Parra-Orobio [1,2], Jonathan Soto-Paz [3], Anulfo Ramos-Santos [2], Keiner Fernando Sanjuan-Quintero [2], Rossember Saldaña-Escorcia [2], Isabel Cristina Dominguez-Rivera [1] and Antoni Sánchez [4,*]

1   Grupo de Investigación en Recurso Hídrico y Saneamiento Ambiental-GPH, Facultad de Ingenierías Fisicomecánicas, Universidad Industrial de Santander, Carrera 27 Calle 9 Ciudad Universitaria, Bucaramanga 680002, Colombia; brayan.parra@correo.uis.edu.co (B.A.P.-O.); isabeldr@uis.edu.co (I.C.D.-R.)
2   Grupo de Investigación Gestión Ambiental y Territorios Sostenibles-GE&TES, Facultad de Ingenierías y Tecnologías, Universidad Popular del Cesar, Seccional Aguachica, Carrera 40 # 1Norte-58, Aguachica 25010, Colombia; aramoss@unicesar.edu.co (A.R.-S.); kfsanjuan@unicesar.edu.co (K.F.S.-Q.); rsaldanae@unicesar.edu.co (R.S.-E.)
3   Grupo de Investigación en Amenazas, Vulnerabilidad y Riesgos a Fenómenos Naturales, Facultad de Ingeniería, Universidad de Investigación y Desarrollo, Calle 9 # 23-55, Bucaramanga 680002, Colombia; jsoto3@udi.edu.co
4   Composting Research Group, Department of Chemical Engineering, Autonomous University of Barcelona, 08193 Barcelona, Spain
*   Correspondence: antoni.sanchez@uab.cat

**Abstract:** The increasing pressure on water resources due to population growth and high water consumption, especially in urban areas from tropical developing countries, has led to a rise in water stress. The sustainability analysis of the Total Water Footprint ($WF_{total}$) and the Environmental Sustainability Index ($SI_{blue}$) are holistic water management tools that allow for establishing pressures over water resources. This study assessed the $WF_{total}$ and their components (Blue, Green, and Gray) in the households of a low-income tropical neighborhood in Colombia with sanitation deficiencies. The activities associated with intra-household water use and higher environmental impact were identified, considering socioeconomic conditions and the water quality in the receiving water body, a wetland, through the application of surveys in a sample of households. The results showed that 86.7% of the WF was the $WF_{green}$, followed by the $WF_{gray}$ (13.2%), and finally, the $WF_{blue}$ (0.2%). The high value of the $WF_{green}$ was related to food consumption patterns, which varied according to socioeconomic level. Likewise, the $SI_{blue}$ shows that the Baquero wetland presented scarcity scenarios regarding water quality and sedimentation, threatening the environmental service provision from this strategic ecosystem. Finally, tools such as the $WF_{total}$ and $SI_{blue}$ help identify strategies that could be implemented to reduce the pressure on the water resources and the water quality degradation in ecosystems relevant to global sustainability as wetlands.

**Keywords:** developing countries; sustainability; water footprint; wastewater; wetlands

## 1. Introduction

In 2018, the proportion of people living in urban areas in the world was 55% and is expected to increase to 68% by 2050 [1]. Industrialization and road connections between cities have promoted urban agglomerations [2] that consume, directly or indirectly, 75% of the global freshwater [3]. This situation increases the pressure on water resources [4] that, exacerbated by climate change, leads to imbalances between water demand and availability, posing significant challenges for water and sanitation provision [5], especially in arid and tropical areas from developing countries [6].

Water is an essential resource, a key element for food, water, and energy security, and is required to achieve the United Nations (UN) Sustainable Development Goals (SDG) [5]. Water significance has been considered explicitly in SDG6, "Clean water and sanitation", which aims to ensure the availability and sustainable management of water and sanitation for all. In addition, water contributes to the goals of SDG2 "Zero Hunger" (food production), SDG7 "Affordable and clean energy" (hydroelectric power generation), and SDG13 "Climate action" (mitigation and adaptation to climate change), among others [7].

Understanding the linkages between productive activities and the pressure over water resources is essential to improve water management and achieve the SDGs. The water footprint (WF) concept is an indicator that allows associating people's consumption practices in specific geographic places with their impact on water resources [8]. The WF estimates the quantity and use of water during a period associated with product manufacturing or service provision. The $WF_{total}$ has three components: (i) Blue water footprint ($WF_{blue}$), the volume of freshwater provided through irrigation and consumed in crop production [9]; (ii) Green water footprint ($WF_{green}$), the amount of rainwater effectively used by crops and irrigation; and (iii) Gray water footprint ($WF_{gray}$) that estimates the amount of polluted water as a consequence of human activities or the amount of water required for dilution of fertilizers and pesticides used in the production processes [10].

WF studies have been carried out mainly in developed countries. Several WF studies have addressed issues such as agricultural water use, climate change, and water for human consumption, considering water as a product and looking to improve water service provision [11]. Another approach for WF studies has been setting standards for cities aiming to undertake transformations toward sustainability. Tiwary et al. [12] assessed India's potential reconfiguration of water demand, reducing the $WF_{gray}$ in water-scarce cities. Islam et al. [3] analyzed WF reduction in multi-regional sectors in Australia, emphasizing that water efficiency programs can reduce $WF_{blue}$ and $WF_{green}$, and contribute to sustainable production planning.

At the interurban scale, Hu et al. [13] found interannual and spatial fluctuations in the WF, allowing a base to support decisions on the sustainable use of water resources. Likewise, Osorio-Tejada et al. [14] highlight the geographical and seasonal variations of the WF, thus, the importance of its contextualization. Finally, Salvador et al. [15] in Spain established, by analyzing water bills, the water requirements of orchards to identify excess irrigation and improve greywater management. However, few studies have used the WF concept in assessing sustainable housing, even more in tropical areas from developing countries, where there are challenges associated with water, sanitation, and hygiene provision [16]. In addition, most of the studies reviewed addressed the city scale [17] and ignored the WF impact of freshwater consumed and used for different purposes over strategic water bodies receiving untreated wastewater discharges [18,19].

Wetlands are strategic ecosystems for climate change mitigation that contribute to biodiversity conservation, flood control, and the provision of a range of ecosystem services to local communities [20]. Since 1971, wetlands have been recognized by the Ramsar Convention as globally important ecosystems, significance ratified by the SDGs [21,22]. The SDG6 promotes the identification and analysis of wetland ecosystem services, including water quantity and quality [22]. Thus, the WF is an indicator that can contribute to quantifying the wetlands' water quality and quantity, generating valuable information for decision-making looking to protect these ecosystems and the services they provide.

The Baquero wetland is in the Magdalena River basin in Colombia. Magdalena is the country's primary water surface source. The Baquero is essential to ensure the well-being of the Magdalena River and the communities that depend on it because of its role as a natural filter that improves river water quality [20]. Additionally, the Baquero contributes to hydrological regulation, preserving biodiversity and supporting people's livelihoods. However, the water use practices of communities around the Baquero adversely affect water availability. This is the case in the Fátima neighborhood (Cesar, Colombia). The Fátima neighborhood had 134 households with 368 inhabitants and is a low-income area

with 100% water supply coverage but a lack of safe wastewater collection and treatment. Thus, Fátima´s untreated wastewater is discharged to the Baquero wetland.

This research was carried out in the Fátima neighborhood and estimates the water footprint and its typologies (WF_blue, WF_green, and WF_gray) according to household practices in a low-income urban area from a tropical developing country and analyzes its impact on a strategic ecosystem receiving untreated wastewater discharges. This work provides information that facilitates identifying anthropic activities and socioeconomic conditions that generate greater pressure on water resources and increases the understanding of the status of wetlands in regions without sanitation. This information allows for identifying contextualized strategies that can be implemented in small tropical cities, contributing to actions toward achieving the targets of the UN SDGs.

## 2. Materials and Methods

### 2.1. Description of the Study Area

The study area is the Fátima neighborhood in the Gamarra municipality (Cesar, Colombia) (Figure 1). Gamarra has 16,644 inhabitants, of which 9377 are in the urban area (56.3% of the population) [23]. The inhabitants of the Fátima are classified as stratum 1 and 2. In Colombia, stratum refers to an economic condition in which stratum 1 has a monthly Gross Domestic Product (GDP) per capita under USD 36 and stratum 6 has a monthly GDP per capita of USD 804 [24]. The neighborhood has an extension of 0.5 hectares and is 50 m above sea level (m.a.s.l.). The climate is warm continental with an average annual temperature of around 28.4 °C, with an annual rainfall of 978 mm concentrated mainly in May, September, and November. People's livelihoods are agriculture, fishing, and livestock keeping. The primary income source is fishing due to the closeness to the Magdalena River (221.6 hectares) and the extension of wetlands (1168.3 hectares).

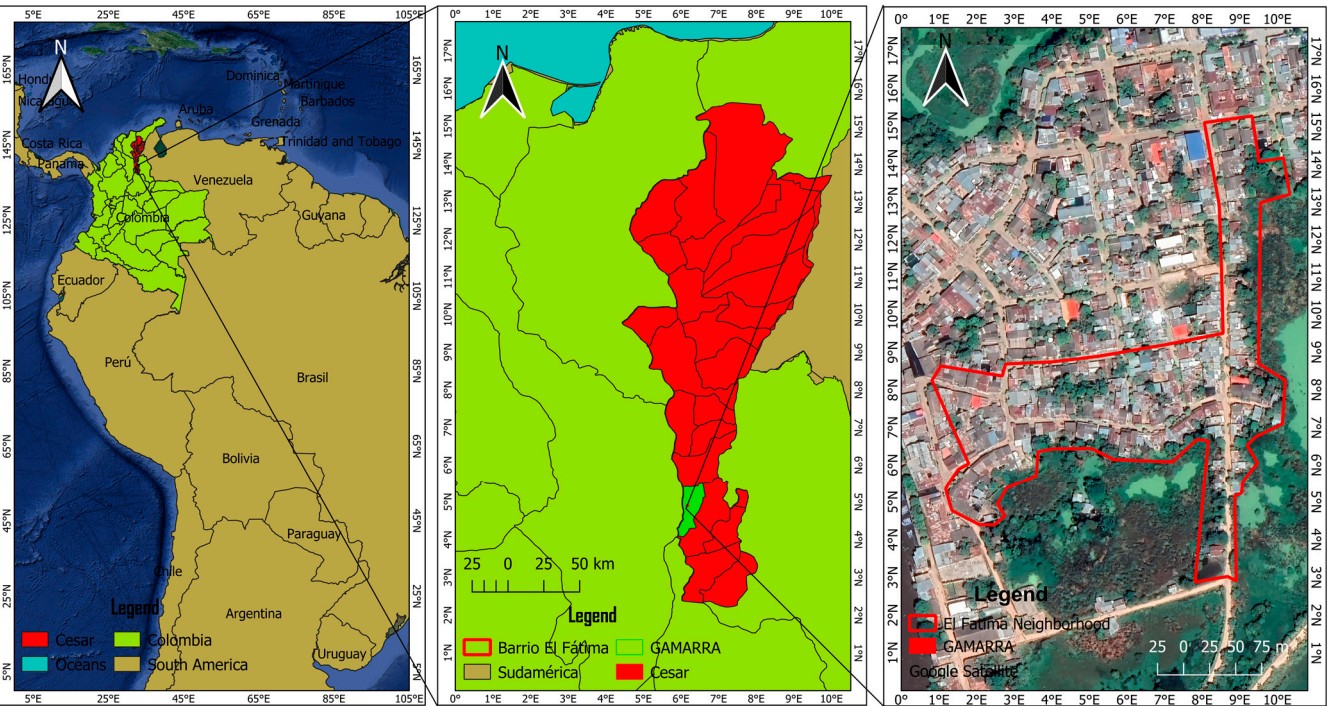

**Figure 1.** Location of the study area: the Fátima neighborhood.

The study area was selected considering: (i) the community´s acceptance to participate in the research, (ii) access to drinking water, and (iii) the environmental impact on the Baquero wetland due to wastewater discharges; the Baquero is the most extensive wetland in the municipality (3204 hectares) and one of the most important wetlands of the Magdalena River. In the last twenty years, population growth has altered the ecological

water quality in the wetland due to the untreated domestic wastewater discharges from this neighborhood (Supplementary Material Figure S1), reducing the ichthyological diversity and other ecosystem services. The demographic and socioeconomic conditions in the study area are common in several developing countries and Latin America. In these countries, as in the study area, water scarcity is expected due to the growing water demand and the lack of collection and treatment infrastructure [25].

On the other hand, the study area had a 100% coverage of water supply, the Magdalena River being the water source. However, there was no water metering or wastewater collection and treatment.

### 2.2. Household Survey

The study area had 368 inhabitants that lived in 137 households. The sample size for the household survey was determined using the equation for sample size to estimate the mean for finite populations with a significance level of 5%, resulting in 101 households to calculate the WF (see the Supplementary Material) [26].

The household survey was prepared, according to recommendations by García and Toro [27], comprising 45 questions (open and multiple-choice), distributed in two sections: (i) socioeconomic aspects and (ii) intra-household water management. The first section included questions regarding the socioeconomic stratum, household size, and food consumption patterns (due to indirect water use associated with food consumption). The second section enquired about personal hygiene, household cleaning activities, and other water uses (car washing and garden irrigation).

The survey was administered through face-to-face interviews by researchers from the Universidad Popular del Cesar, Sectional Aguachica, targeting adult household members. The interviewers carried the University Identification Card and explained the project objectives in each selected household before applying the survey. Interviewees granted informed consent before answering the questionnaire. Surveys were conducted from 8:00 a.m. to 12:00 p.m. and 2:00 p.m. to 5:00 p.m. in July 2022.

### 2.3. Water Footprint Estimation

The WF can be assessed by considering the activities developed in a specific region. In addition, the WF can be estimated from two perspectives: production and consumption [28]. This research is focused on the WF associated with the direct use of water from a specific region, corresponding to the freshwater consumed in consolidated urban areas (i.e., urban areas with more than 50% built [29]). The three components of the WF were estimated: $WF_{blue}$, $WF_{gray}$, and $WF_{green}$, and a systematic tool for the WF estimation was developed. This tool also identifies the activities with higher WF in the study area.

The $WF_{blue}$ in urban areas, according to Manzardo et al. [30], represents the fraction of freshwater that evaporates from different sources (roads, rivers, lakes, etc.), including water consumed by communities resulting from their typical activities or lost in different processes (refrigeration, transport, heating, storage, etc.). In this case, $WF_{blue}$ was obtained through the volumetric data of water used for drinking and cooking collected through the household survey (Equation (1)).

$$WF_{blue} = V * month \tag{1}$$

$WF_{blue}$ is the Blue Water Footprint (L/month), V is the amount of water used for daily drinking and cooking, and the month is the equivalent of 30 days.

$WF_{gray}$ is the total volume of freshwater used to assimilate the contaminant load discharged during the production of products and services. The $WF_{gray}$ allows for identifying water scarcity problems associated with water quality and anthropic activities' impact on water resources at different spatial scales [29]. For the $WF_{gray}$ calculation, typical daily

activities such as showering, brushing teeth, and washing dishes, among others, were assessed (see Equation (2)).

$$WF_{gray} = F_{act} * V_{act} * Inhab * month \tag{2}$$

where $WF_{gray}$ is the Gray Water Footprint (L/month), $F_{act}$ in the daily frequency of the activity, $V_{act}$ is the amount of water consumed by the activity (L), Inhab is the number of inhabitants that develop the activity, and a month is equivalent to 30 days. Likewise, $WF_{gray}$ considered the wastewater discharges ($WF_{gray-D}$) from the study area to the Baquero wetland (Equation (3)). For this, six discharge points over the Baquero wetland were selected and analyzed for parameters such as the Chemical Oxygen Demand (COD) and Total Suspended Solids (TSS), following APHA standards [31]; these parameters were selected according to recommendations from previous studies [8]. The selected six points were the most critical pollution points identified from information provided by the environmental authority in the study area (CORPOCESAR) through interviews with neighborhood social leaders (supplementary material Figure S1).

$$WF_{gray-D} = \frac{L_{COD}}{C_{maxCOD} - C_{rbCOD}} + \frac{L_{TSS}}{C_{maxTSS} - C_{rbTSS}} \tag{3}$$

where $L_{COD}$ is the COD load at the discharge point (kg/s), $L_{TSS}$ is the TSS load at the discharge point (kg/s), $C_{maxCOD}$ is the maximum allowable concentration of COD in the receiving body (kg/m$^3$), $C_{maxTSS}$ is the maximum allowable concentration of TSS in the receiving body (kg/m$^3$), $C_{rbCOD}$ is the COD concentration in the receiving body (kg/m$^3$), and $C_{rbTSS}$ is the TSS concentration in the receiving body. The maximum allowable concentrations were established following the Colombian wastewater discharge standards (Resolution 631 of 2015).

The $WF_{green}$ is indirectly related to anthropic activities, mainly evapotranspiration from the forest and agricultural fields. In this study, the $WF_{green}$ was estimated from the information on the product purchases of the family basket acquired at a particular time (daily, weekly, or monthly).

$$WF_{green} = P_p * WF_{foodi} * mont \tag{4}$$

where $WF_{green}$ is the Green Water Footprint (L/month), $P_p$ is the vegetal or animal product purchased for the week (kg or L in units), $WF_{foodi}$ is the quantity of product i (vegetal or animal food) (kg, L or quantity in units) and month is the equivalence to four weeks.

Finally, $WF_{total}$ is the summation of $WF_{blue}$, $WF_{green}$, and $WF_{gray}$.

$$WF_{total} = WF_{blue} + WF_{green} + WF_{gray} \tag{5}$$

### 2.4. Activities with Higher WF Demand

The population's water-use expectations are fundamental to estimating the WF [32]. These expectations were captured through the household survey previously described. A list of the actions expressed by the interviewed people contributing to the WF in the studied area was prepared. Causes and consequences around the study's central problem (pressure over water resources) were also included.

The Vester matrix [33] was prepared to identify and analyze causes, effects, and central problems, directly and indirectly, generating pressure over water resources in the study area. In addition, this matrix relates the dependency of problems on a 0–3 scale, where 0 is no direct relationship between both problems and 3 is a strong influence. The matrix prioritizes problems by determining causes and effects according to the location in a Cartesian plane. Then, each problem is classified (passive, critical, indifferent, and active) and is located in the plane based on the results of the summation of rows (causal influence (X)) and columns (dependency (Y)) [33].

### 2.5. Sustainability Assessment of the Water Footprint Using Indicators

Sustainability indicators allow users (general public, researchers, and policymakers) to collect information in a simple and quantifiable fashion that helps make decisions that facilitate managerial or governance processes at different scales [34]. According to Michalina et al. [35], indicators are recommended and effective in assessing a region's urban development sustainability. Likewise, indicators focused on sustainable development are used as primary sources of information in structuring and formulating prioritized strategies, programs, and policies aiming to address a relevant problem [36].

The Blue Water Scarcity Index ($SI_{blue}$) (Equation (6)) is the relation between the summation of the Blue Water Footprint ($\sum WF_{blue}$) and the Blue Water Availability ($Av_{blue}$) [37]. This indicator seeks to associate the amount of water for consumption. The Water Pollution Level ($WPL_{gray}$) (Equation (7)) is the relation between the summation of the Gray Water Footprint ($\sum WF_{gray}$) in the catchment area and the $Av_{blue}$, and aims to reflect the local impact of the environmental water quality in a region [38].

$$SI_{blue} = \frac{\Sigma WF_{blue}}{Av_{blue}} \tag{6}$$

$$WPL_{gray} = \frac{\Sigma WF_{gray}}{Av_{blue}} \tag{7}$$

According to Hoekstra et al. [29], the $Av_{blue}$ could be 20% of the runoff in the catchment area which, in this case, is the Magdalena River, due to its influence in the study area. The data on the behavior of the river corresponded to one year according to the reports of the Colombian authority for these matters, which is the IDEAM (Institute of Hydrology, Meteorology and Environmental Studies). Likewise, the values recommended by Hoekstra et al. [29] were considered to assess the indices, as shown in Table 1.

**Table 1.** Assessment range.

| Index | Range | WF$_{blue}$ | WF$_{gray}$ |
|:---:|:---:|:---:|:---:|
| A | <1.0 | Very low water scarcity | Very low water pollution index |
| B | 1.0–1.5 | Moderate water scarcity | Moderate water pollution index |
| C | 1.5–2.0 | Significant water scarcity | Significant water pollution index |
| D | >2.0 | Severe water scarcity | Severe water pollution index |

Adapted from Hoekstra et al. [34].

### 2.6. Statistical Analysis

The statistical analysis was performed using the free distribution software R version 3.5.1. Statistically significant differences and relevant correlations between the parameters at a significance level of 5% were established [26]. The relationship between WFs and their components was assessed using a multivariate Principal Component Analysis (PCA). The principal components were those with an eigenvalue greater than one and statistically significant according to the parsimony principle. In addition, to validate the PCA, a Kaiser–Meyer–Olkin index higher than 0.5 was required [39]. The strength of the linear relationship between the WFs was determined through the Pearson coefficient (R) and the *p*-value at a significance level of 5%. In this study, a relationship was considered to be strong when the value of R was greater than or equal to 80%. Therefore, the correlation is moderate to weak when R is lower than 80% [40]. Finally, a hierarchical grouping (Ward's method) was used to determine possible clusters between the WFs, looking for a minimum variance between the clusters [41]. Information processing was developed using the SPSS®Statistics 25.0.

## 3. Results

### 3.1. Water Footprint Estimation

According to the survey results, 52% of the population was in stratum 1, and 48% was in stratum 2 (Supplementary Material Figure S2). This population pattern is typical in urban

areas such as the study area [23]. In stratum 2, there were more children than in stratum 1, 34% compared to 29% (see Supplementary Material Figure S3). This behavior contrasts with findings from Charles-Coll et al. [42], who indicate that in developing countries, as families increase their purchasing power, they tend to have fewer children. This atypical behavior can be related to local culture since Gamarra city is on the Atlantic coast of Colombia, where household size increases with the economic level [43].

Table 2 shows the Water Footprint results and its categories (Blue, Green, and Gray) for the Fátima. During the studied period, the neighborhood had a $WF_{total}$ of 26,213.7 m$^3$/month, where 86.7% was $WF_{green}$ (water consumed in food production), 13.2% was $WF_{gray}$ (water used for daily activities such as showering, brushing teeth, washing dishes, among others), and 0.2% was $WF_{blue}$ (water used for drinking and cooking). These results are similar to those reported by Hirpa et al. [44] in Ethiopia, who found that more than 85% of the WF was in the green category.

**Table 2.** Water Footprint of the Fátima neighborhood.

| WF Category | (m$^3$/Month) |
|---|---|
| Blue Water Footprint | 48.0 |
| Green Water Footprint | 22,717.1 |
| Gray Water Footprint | 3448.5 |
| Total Water Footprint | 26,213.7 |

Regarding the amount of water consumed, the per capita water consumption was 9.9 m$^3$/month and per family 41.7 m$^3$/month (household size of four people). These results are higher compared to average values in Colombia, both per capita (3.8 m$^3$/month) and per household (15.4 m$^3$/month) [45]. Likewise, these results differ from a WF assessment in Distrito de Chorrillos in Perú, for a sample of 368 people (similar to the studied population and with an average temperature of 23 °C) where the $WF_{total}$ was 39,023.9 m$^3$/month [46]. The results from Chorrillos are not ascribed to best water-managerial practices but to insufficient access to water due to political–economical causes [46]. On the other hand, the results show similarities with Carrascal and Londoño [47] in Buenavista-Córdoba, Colombia (average temperature: 28 °C and similar socioeconomic conditions than our study area), who found that, in the average household, $WF_{blue}$ and $WF_{gray}$ were 16.5 m$^3$/month, while for the Fátima it was 34.6 m$^3$/month.

Figure 2 shows the $WF_{blue}$ distribution according to consumption and the strata in the study area (1 and 2). $WF_{blue}$ was 48.0 m$^3$/month, with stratum 1 having the superior value (27 m$^3$/month). $WF_{blue}$ for stratum 2 was 21.1 m$^3$/month.

On average, the water consumed for drinking and cooking by the household members in the neighborhood, excluding pets, was 8.9 m$^3$/month, equivalent to 74.2 Lpcd (Litres per capita per day), probably associated with the altitude of the study area, since the Gamarra municipality is over 50 masl (meters above sea level), with an average temperature that favors dehydration. This value agrees with reports by Manco-Silva et al. [48], who highlight that the increase in potable water consumption is associated with maximum temperatures and the number of days with rainfall.

Regarding households in stratum 2, a lower $WF_{blue}$ (43.9%) than in stratum 1 was identified (56.2%). In stratum 1, water demand for food preparation was higher (15.9 m$^3$/month) compared to stratum 2 (12.6 m$^3$/month). This was, on average, 10.1 Lpcd for stratum 1 and 8.8 Lpcd for stratum 2. Both strata surpassed the values recommended by the World Health Organization (WHO)—6.5 Lpcd to ensure drinking and food preparation [49]. This behavior is linked to purchasing power since households in stratum 2 own items such as air conditioning and have better construction characteristics (ceramic floor, ceiling, among others). These aspects could contribute to a reduction in water consumption compared to stratum 1. Authors such as Manco-Silva et al. [48] and Hidalgo et al. [50] highlight the relationship between household size, socioeconomic level, and water-efficient practices in the household.

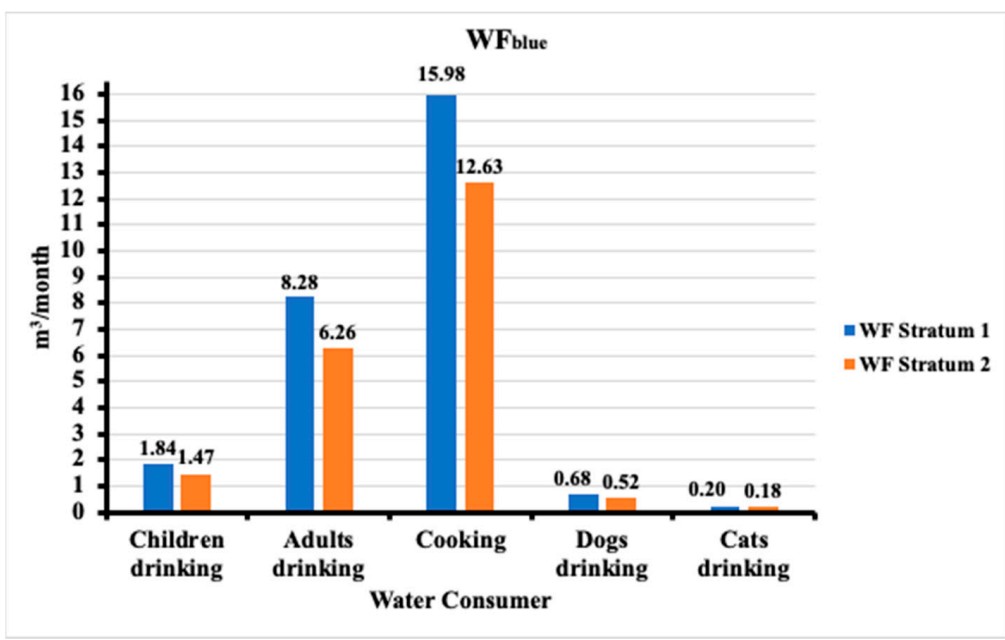

**Figure 2.** Distribution of the WF$_{blue}$ in the Fátima neighborhood according to water consumer and socioeconomic stratum.

In addition, the behavior in stratum 1 could also be associated with hydraulic deficiencies in wastewater management facilities. Even though the area lacked a wastewater collection system, a higher quantity of water was required to evacuate excreta to the Baquero wetland since the designs of these structures did not meet the optimal conditions for evacuating this wastewater. Furthermore, the lack of water metering and the low water tariffs in the municipality could favor the high-water domestic consumption found compared with Colombian averages. Water company employees also identified this situation through visits made in previous years, where the local people expressed low interest in water reuse. These aspects could be associated with the little valuation of water conservation and the impacts on water-related ecosystems such as the wetland, similar to findings from San Cristobal in Venezuela [51].

WF$_{green}$ was 22,717.1 m$^3$/month, higher in stratum 2 (11,622.5 m$^3$/month) compared to stratum 1 (11,094.6 m$^3$/month), a 4.5% difference among strata (527.9 m$^3$/month). Figure 3 shows the WF$_{green}$ distribution according to food consumed.

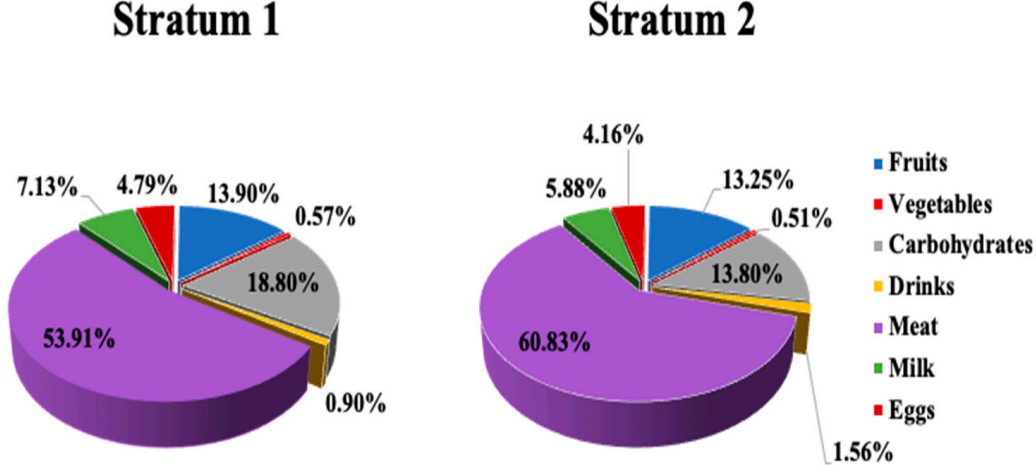

**Figure 3.** Distribution of WF$_{green}$ according to the type of food and stratum in the Fátima neighborhood.



Animal protein (beef, pork, fish, and chicken) was the type of food with a higher incidence in $WF_{green}$ in stratum 1, comprising 53.9% (5981.1 $m^3$/month), beef having the greatest impact (2133 $m^3$/month) since the production of 1 kg of beef demands approximately 15,000 L of water [52]. People in stratum 1 bought, on average, 313.4 kg of animal protein per week, fish being the protein source with highest consumption (3.2 kg/person-week), followed by chicken (1.9 kg/person-week), pork (1.4 kg/person-week), and beef (1.2 kg/person-week). Although in the study area fish, chicken, and pork were eaten in greater quantities than beef, their WF was lower due to their lower water requirements to produce 1 kg compared to the beef demand (3300 L, 4325 L, and 5988 L, respectively) [52,53]. The behavior found regarding fish consumption occurred because most of the population were fishers in the Magdalena River and the Baquero wetland. Moreover, people lacked the economic capacity to buy other animal proteins.

On the other hand, 18.8% of $WF_{green}$ in stratum 1 was carbohydrates consumption (kg/household-week), where rice was the most popular (3.3), followed by maize (2.40), sugar (2.3), potato (2.0), cassava (1.5) and wheat (1.0). Rice had a $WF_{green}$ of 1180.1 $m^3$/month, while maize had 379.6 $m^3$/month. This behavior agrees with reports from Das et al. [54], who indicate that rice typically has the highest water demand among carbohydrates globally. Fruit consumption contributed 13.9% of the $WF_{green}$, plantain having the greatest demand for all the inhabitants in stratum 1 (3.1 kg/home-week), equivalent to a monthly water demand of 966.1 $m^3$.

Stratum 2 contributed higher to the $WF_{green}$ in the study area, as in stratum 2, consumption of animal protein was 27 kg/home above stratum 1, with the same tendency concerning the preference of proteins in kg/inhabitant-week: fish (2.7), chicken (2.3), pork (1.9), and beef (1.3).

In general, the socioeconomic level influences $WF_{green}$, and as the stratum increases, the population accesses a more diverse diet, increasing the consumption of animal protein, which increases the $WF_{total}$. In contrast, a diet with carbohydrate predominance and less animal protein reduces water consumption. Tuninetti et al. [55] indicate that a healthy carbohydrate-rich diet reduces pressure over water resources and, thus, reduces the WF.

The $WF_{gray}$ is related to the daily activities that impair the physicochemical water quality without wastewater treatment and generate environmental problems due to pollutants and excess nutrients. This situation affects aquatic ecosystems due to the proliferation of species such as invasive macrophyte plants and the increase of sediments that reduce navigation and artisanal fishing, which is an essential people´s livelihood for many low-income communities [8,56]. All these issues are present in the study area due to untreated wastewater discharges to the Baquero wetland.

In the study area, the activities that contribute to the $WF_{gray}$ (3448.5 $m^3$/month) were distributed as shown in Figure 4: personal hygiene that includes brushing teeth, handwashing, face washing, shaving, and showering with 2002.9 $m^3$/month; cleaning that includes clothes washing, washing dishes and general household cleaning with 1425.8 $m^3$/month; and other consumption (garden irrigation and car washing) with 19.8 $m^3$/month.

Regarding personal hygiene, water consumption for showering had a higher demand, with stratum 2 having a lower demand (615.1 $m^3$/month) than stratum 1 (783.7 $m^3$/month). This result in stratum 2 could be explained by the lower population (177 inhabitants from 368) and higher number of households with showers (40 households from 48) than in stratum 1. The contributions of personal hygiene activities to the $WF_{gray}$ were in order: toilet flushing (340.7 $m^3$/month), handwashing (119.7 $m^3$/month), and face washing (87.7 $m^3$/month).

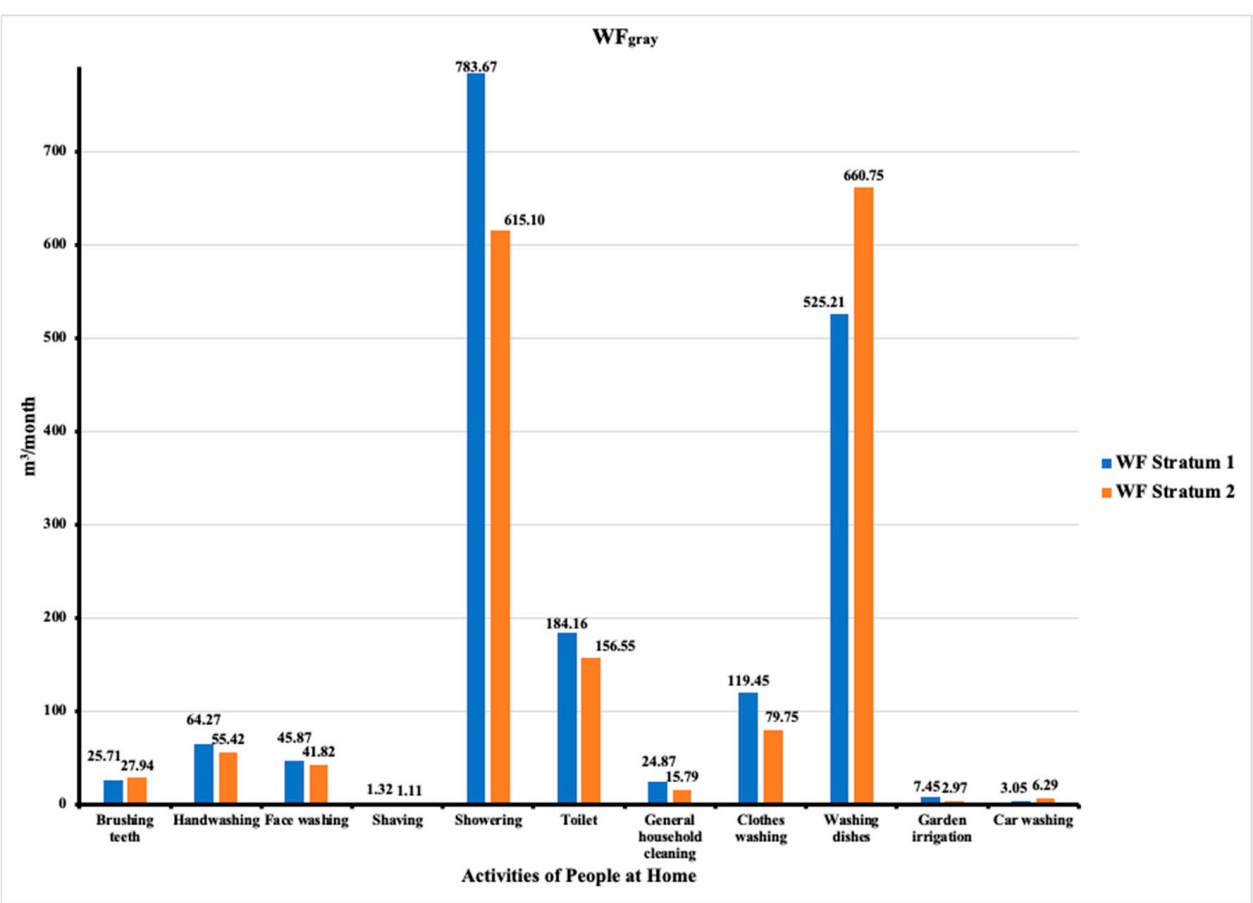

**Figure 4.** Gray Water Footprint from daily activities according to socioeconomic stratum in the Fátima neighborhood.

Concerning activities associated with household cleaning, stratum 2 consumed 660.8 m³/month in washing dishes, higher than the consumption in stratum 1 (525.2 m³/month), possibly associated with a longer duration of this activity evidenced by the household survey. Water wastage in household cleaning could be explained due to the low water tariff, which leads to a lack of recognition of the total water value (benefits) that prevents the adoption of water conservation practices [51,57].

On the other hand, people from stratum 1 wash clothes more frequently (46 times per week), which leads to an average consumption of 96.1 L per washing cycle, different from stratum 2 (34 times a week), 86.5 L per washing cycle. The clothes-washing frequency could be linked with local habits identified through the household survey. For instance, some people washed their clothes constantly, not once a week. In addition, household cleaning in stratum 1 also had higher WF than stratum 2 due to a higher weekly frequency, five times for stratum 1 and four times for stratum 2.

Concerning other water consumptions, garden irrigation was an activity with higher demand than car washing since most people from stratum 1 also threw water into the streets to minimize the dust raised by the wind (roads were paved only in the stratum 2 sector). On the other hand, in stratum 2, there were 27 cars and 18 in stratum 1. Thus, the difference in water consumption for car washing between strata was 3.2 m³/month. Household car washing is forbidden in Colombia (Law 1801 of 2016).

Concerning $WF_{gray-D}$, the study area produced 4768.8 m³/month, of which 3308.8 m³/month were associated with TSS and 1460 m³/month with COD. The important flow of solids entering the wetland from the discharge of untreated wastewater from the study area is reflected in the reduction of the wetland surface area and the increase in sediments that reduce aquatic life. This situation has not been previously reported in this

strategic ecosystem of the Magdalena River. On the other hand, the input of organic load identified in this research is low compared to other studies such as those developed by Contreras and Torres [58] (5153.5 m$^3$/month) in Monteria, Colombia and Vieira and Junior [59] (6.8 × 10$^6$ m$^3$/month) in Caraguatatuba, Brazil. These studies' differences can be attributed to the population, wastewater collection systems, and economic activities. In addition, the studies were developed in urban areas with populations above 100,000 inhabitants, contrasting with the fewer than 10,000 people that the urban area of Gamarra had.

In this case, the WF$_{gray-D}$ found indicates that effluents from the studied area did not meet the national water quality standards. It also shows that the water quality from the wetland cannot dilute the wastewater inputs from the neighborhood. According to Hoekstra et al. [29], if the WF$_{gray}$ is lower than the flow of the water-receiving body, there is still enough water to dilute pollutants at a concentration below the current standards (Resolution 631/2015). The WF$_{gray-D}$ for wastewater discharges from the study area (3448.5 m$^3$/month and 4768.8 m$^3$/month, respectively) did not exceed the annual water offer in the dry year for the Magdalena River, which is hydrologically connected with the Baquero wetland (27,338 Mm$^3$) [60]. However, this is not the only community discharging untreated wastewater to the Magdalena River. Therefore, studies are needed to address the pollution due to the wastewater contributions to the wetland since eutrophication and sedimentation are already present [56].

Finally, statistically significant differences in WF$_{blue}$ and WF$_{gray}$ were found between socioeconomic strata ($p < 0.05$) (see Supplementary material), showing that the average water consumption was not different between strata but different due to the activities in each stratum. A post-ANOVA allowed to identify that, among the socioeconomic strata, there were statistically significant differences only for the activities related to showering and washing dishes. This result agrees with findings from Mazzoni et al. [61], who argue that showering could be up to 38% of the total water consumption in a household (average of 121.5 Lpcd). Likewise, Richter and Stamminger [62] found that dishwashing could generate a substantial water loss of up to 58% of the water demand for activities related to the kitchen in households.

### 3.2. Principal Component Analysis of Water Footprints

For each type of water footprint, the PCA presented eigenvalue values greater than one. In addition, the two first principal components (F1 and F2) explained more than 50% of the data variability. Figure 5A shows the PCA of the water footprints. The Pearson correlation matrices (r) are presented in Supplementary Material (Tables S2–S5).

The PCA showed that the estimated WFs inside the households were related. However, three clusters were identified (Figure 5A): (i) Blue cluster with the WF$_{blue}$ and WF$_{green}$; (ii) Grey cluster with WF$_{gray}$, and (iii) WF$_{gray-others}$. The results suggest that WF$_{blue}$ is related to WF$_{green}$ due to the transfer of intersectoral virtual water from producers to consumers [39]. These results are similar to those reported by Chai et al. [39], who highlight that households use more water indirectly as virtual water incorporated into their daily consumption due to crop production.

On the other hand, WF$_{blue}$ and WF$_{green}$ have a positive correlation with the WF$_{gray}$; therefore, an increase in one generates an increase in the other. Thus, the water used in the Fátima generates wastewater that can pollute the Baquero wetland. In contrast, the WF$_{gray-others}$ was located in the second quadrant of the PCA, indicating a low relationship with the other water footprints (R < 0.5).

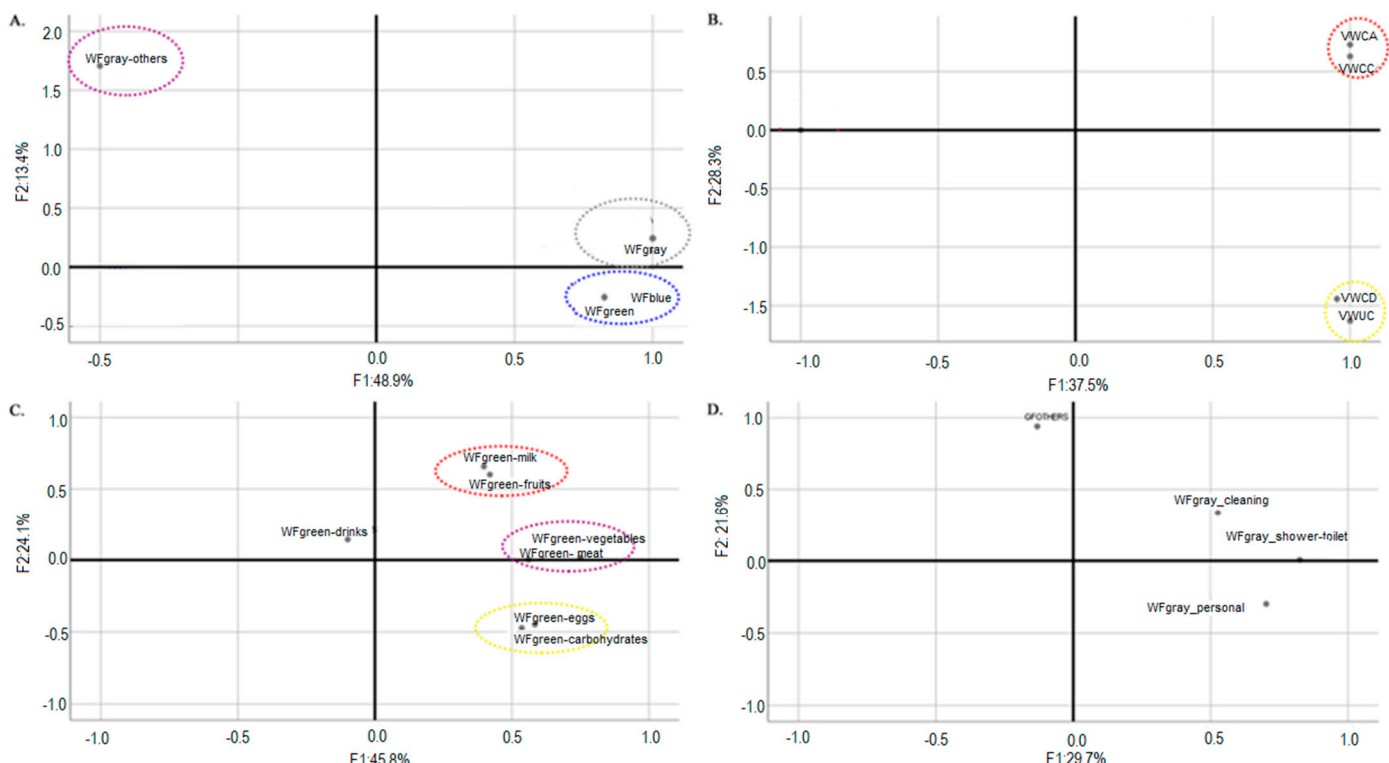

**Figure 5.** (**A**). Principal Component Analysis of the Water Footprints estimated for the Fátima neighborhood. (**B**). PCA of the elements that constitute the $WF_{blue}$ (**C**). PCA of the elements that constitute the $WF_{green}$. (**D**). PCA of the elements that constitute the $WF_{gray}$.

Figure 5B shows the distribution of the different $WF_{blue}$ components in the loading plot. The volume of water consumed by adults (VWCA) and the volume of water consumed by children (VWCC) was in the upper right quadrant (red cluster). The proximity between the two volumes is associated with the water requirements for drinking and hygiene, which are influenced by the region's average temperature (28.4 °C). In this regard, Yan [63] showed that domestic water consumption in Urumqi, China was highly correlated with temperature and weakly correlated with rainfall. Results indicate that VWCA and the volume of water consumed for cooking (VWCCo) were the main contributors to the $WF_{gray}$ in the households. In contrast, the lower contribution to the $WF_{blue}$ was the water consumed by dogs (VWCD) and cats (VWUC), located in the lower right quadrant (yellow cluster).

Figure 5C shows the distribution of the elements constituting the $WF_{green}$ in the PCA. Three clusters are prominent: (i) the footprint generated by eggs and carbohydrates (yellow cluster); (ii) milk and fruits (red cluster); (iii) and meat and vegetables (purple cluster). Meat and vegetables were the major contributors to the WF in the study area, evidenced by a water consumption of 1050.7 m$^3$/month. Likewise, these products were the major contributors to the $WF_{green}$. On the other hand, fruits and dairy products were close to the F1 component, showing a lower consumption compared to the violet cluster. This result can be explained because in the food family basket, fruits such as plantain, bananas, and guava were the most consumed, and their water demand per kg produced was: 1.6 L, 1.4 L, and 1.5 L, respectively. Regarding the red cluster, $WF_{dairy}$ and $WF_{fruits}$ were characterized by having the lowest contribution to the green water footprint despite being positively correlated to the other elements that make up the $WF_{green}$ (see Supplementary Material).

Regarding the $WF_{gray}$, made of the daily activities that impair water's physicochemical and biologic characteristics, no defined clusters resulted. The observed scatter pattern could be associated with each household's cultural practices [64]. Gregory and Di Leo [65] indicate that water management devices and conservation practices influence intrahousehold water use. The $WF_{gray}$ found in this research differs from findings in India and Nigeria, where

water consumption is limited due to availability and supply [66], but lower compared with values from Hong Kong and Beijing, where there is a direct relationship between the Gross Domestic Product (GDP) and the increase on the base water use rate [67].

The water consumption associated with the different footprints in the study area demands the integration of synergistic strategies aimed at reducing household water consumption, addressing water scarcity, and pollution control [68] such as: (i) awareness and education campaigns, that communicate information on how to reduce water demand, how to use water more efficiently, and how to reuse water; (ii) programs for the installation of water-saving devices, that include financial incentives, such as discounts on water bills or low-interest financing programs; (iii) regulations on appliance efficiency, involving regulating the efficiency of washing machines; (iv) rainwater harvesting programs, that include incentives for the installation of rainwater harvesting systems and the use of collected water for non-potable activities, such as garden irrigation; (v) promotion of water-reuse technologies, including financial incentives and education on the implementation of greywater reuse systems; and (vi) sustainable food consumption and production, that encourage food waste minimization and efficient cooking methods [69], since food consumption patterns influence environmental pressures, and behavioral change in this matter provides opportunities to alleviate the current water stress in some regions [70]. These strategies must be assessed within an integrated framework to prevent adverse effects and control the "rebound effect," which can occur when a measure intended to save water increases water consumption [71].

The estimation of the WF in the context studied implied methodological adaptations to overcome information gaps commonly existing in developing countries, for example, identifying people's food consumption habits and intrahousehold water consumption patterns in an area without micrometers. This lack of information was solved with the household survey. Thus, this study can help researchers and decision-makers in similar contexts to carry out WF studies that help identify strategies to reduce pressure on water resources according to socioeconomic conditions. Greater adoption of this methodology in these contexts facilitates understanding the interaction between social relations and economic and natural resources and would improve water management, as in developed countries [72].

### 3.3. Identification of the Activities with the Greatest Demand of Water Footprint and Environmental Sustainability Index

The Vester matrix allows the identification of the active problems that cause the high water consumption in the households of the Fátima. The study area's water company had not previously systematically identified this situation's causes. Table 3 presents the valuation of each problem, and Figure 6 includes the Cartesian plane that illustrates the relationship between problems.

As a result of the Vester matrix and the Cartesian plane, each factor's degree of incidence concerning the problem raised in the matrix was identified. Indifferent problems were: P1: Leaving the tap running when taking a shower; P2: Leaving the tap running when brushing teeth; P4: Leaving the tap running when washing dishes; P8: Excess of clothes washing; P12: Lack of inspection and maintenance of household pipes, and P13: Lack of rainwater harvesting. Passive problems were: P3: Use large amounts of water to reduce dust in the streets and P9: Lack of water reuse. On the other hand, the most critical problem was: P6: Irrational water use. Finally, the active problems were: P5: Lack of water-saving devices; P7: Lack of sectorization by the water company; P10: Large household size, and P11: Lack of household water metering.

**Table 3.** Vester matrix for the high-water consumption in households from the Fátima neighborhood.

| Code | Variable | P1 | P2 | P3 | P4 | P5 | P6 | P7 | P8 | P9 | P10 | P11 | P12 | P13 | Total Active |
|------|----------|----|----|----|----|----|----|----|----|----|-----|-----|-----|-----|--------------|
| P1 | Leaving the tap running when taking a shower. | | 0 | 0 | 0 | 0 | 3 | 0 | 0 | 3 | 0 | 0 | 0 | 0 | 6 |
| P2 | Leaving the tap running when brushing teeth. | 0 | | 0 | 0 | 0 | 3 | 0 | 0 | 3 | 0 | 0 | 0 | 0 | 6 |
| P3 | Use large amounts of water to reduce dust in the streets. | 0 | 0 | | 0 | 0 | 3 | 0 | 0 | 2 | 0 | 0 | 0 | 0 | 5 |
| P4 | Leaving the tap running when washing dishes. | 0 | 0 | 0 | | 0 | 3 | 0 | 0 | 3 | 0 | 0 | 0 | 0 | 6 |
| P5 | Lack of water-saving devices. | 3 | 3 | 3 | 3 | | 3 | 0 | 0 | 0 | 0 | 0 | 0 | 0 | 15 |
| P6 | Irrational water use. | 3 | 3 | 3 | 3 | 0 | | 0 | 3 | 3 | 0 | 0 | 0 | 0 | 18 |
| P7 | Lack of sectorization by the water company. | 3 | 3 | 3 | 3 | 0 | 3 | | 3 | 3 | 0 | 0 | 0 | 3 | 24 |
| P8 | Excess of clothes washing. | 0 | 0 | 0 | 0 | 0 | 3 | 0 | | 3 | 0 | 0 | 0 | 0 | 6 |
| P9 | Lack of water reuse. | 0 | 0 | 2 | 1 | 0 | 3 | 0 | 1 | | 0 | 0 | 0 | 3 | 10 |
| P10 | Large household size. | 2 | 2 | 1 | 1 | 0 | 3 | 0 | 2 | 1 | | 0 | 0 | 2 | 14 |
| P11 | Lack of household water metering. | 3 | 3 | 3 | 3 | 3 | 3 | 0 | 3 | 2 | 0 | | 2 | 1 | 26 |
| P12 | Lack of inspection and maintenance of household pipes. | 0 | 0 | 0 | 0 | 0 | 1 | 0 | 0 | 1 | 0 | 0 | | 0 | 2 |
| P13 | Lack of rainwater harvesting. | 0 | 0 | 2 | 0 | 0 | 2 | 0 | 1 | 2 | 0 | 0 | 0 | | 7 |
| | **Total passive** | 14 | 14 | 17 | 14 | 3 | 33 | 0 | 13 | 26 | 0 | 0 | 2 | 9 | 145 |

Note: P stands for problems.

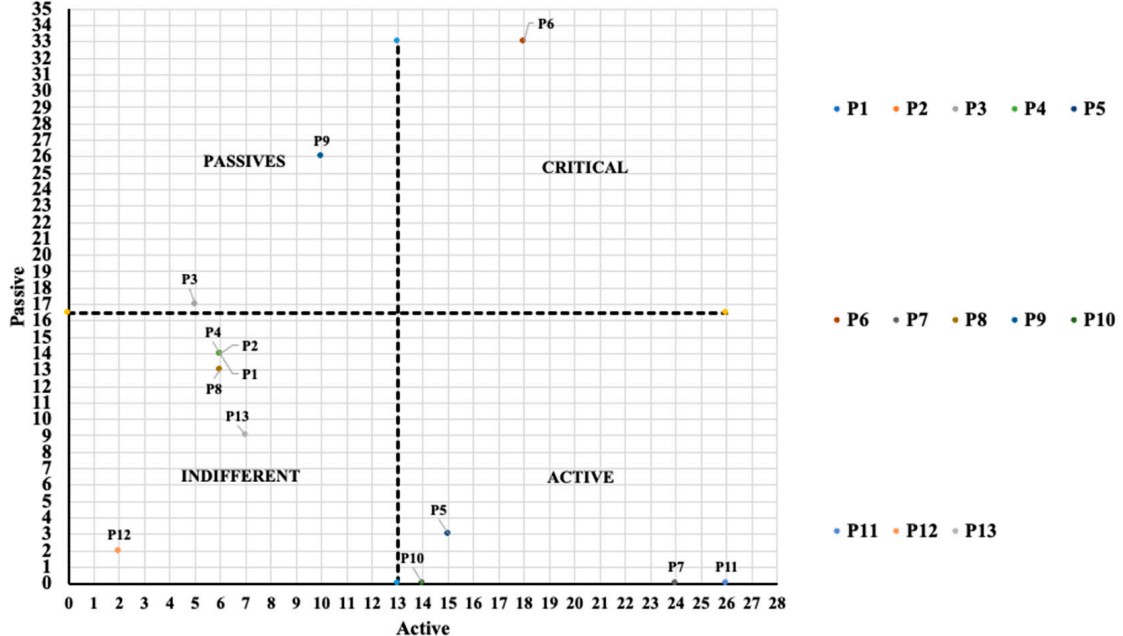

**Figure 6.** Cartesian plane of high water consumption in the households of the Fátima neighborhood.
Note: P stands for problems.

Water wastage was the most notorious problem in the study area. These results are consistent with those reported by Bellot and Fiscarelli [73] and Murwirapachena [74], who indicate that the change in consumption habits to efficient water use practices and the implementation of efficient water devices could lead to substantial reductions in domestic water demand. Thus, actions such as those recommended by Garcia et al. [75], including household metering, implementing low water consumption devices (flow reducers, mixer-aerators, and sliders-volume regulators), and environmental education could promote changes in water consumption patterns at the household level and attitudes for valuing water. These strategies could greatly impact reducing pressure on water resources in an area vulnerable to stress and water insecurity exacerbated by climate change.

It was found that 50% of the active problems were related to the water service provider in the Gamarra municipality, associated with the lack of water metering, favoring the absence of water conservation and saving practices. Although in urban areas of Colombia, the recommendations of water companies about the rational use of water resources are common, this is predominantly in large cities with economic resources. At the same time, small municipalities such as Gamarra typically lack resources to develop educational and communication strategies around water [76]. Therefore, this shows institutional weaknesses and challenges for water governance in the area, causing difficulties in meeting the SDGs, particularly SDG6.

Tables 4 and 5 show the results of the Environmental Sustainability Indices in the Fátima neighborhood.

According to Tables 4 and 5, in the period addressed in this research, the Magdalena River could supply the water demand (quantity) from the community through the water supply system. However, this water consumption leads to a non-sustainability situation regarding the $WF_{gray}$. This situation is because the Water Pollution Level ($WPL_{gray}$) was higher than 2.0, being in a severe scarcity category. In addition, more than 40% of the water available in the receiving body was consumed, which implies that the ecological flow was not guaranteed and that the water source used had a quality that was increasingly deteriorating. The situation described is a common phenomenon in developing countries in Latin America, Africa, and Asia, where the pollution of several rivers is critical [17]. Colombia follows this pattern since approximately 50% of the country's water sources are categorized as poor quality [77].

**Table 4.** Environmental Sustainability for the Blue Water Scarcity Index ($SI_{blue}$).

| Month | Average Monthly Flow Magdalena River (m$^3$) | Monthly Flow (m$^3$) | $Av_{blue}$ (m$^3$) | $SI_{blue}$ | Sustainability |
|---|---|---|---|---|---|
| January | 1998.9 | 59,965.7 | 354.7 | 0.135 | Very low scarcity |
| February | 2001.9 | 60,056.9 | 355.2 | 0.135 | Very low scarcity |
| March | 3367.6 | 101,028.0 | 597.5 | 0.080 | Very low scarcity |
| April | 4619.3 | 138,580.2 | 819.6 | 0.059 | Very low scarcity |
| May | 4831.9 | 144,959.4 | 857.4 | 0.056 | Very low scarcity |
| June | —— | —— | —— | —— | —— |
| July | 3940.3 | 118,208.5 | 699.1 | 0.069 | Very low scarcity |
| August | 4004.9 | 120,145.5 | 710.6 | 0.068 | Very low scarcity |
| September | 3672.7 | 110,181,6 | 651.7 | 0.074 | Very low scarcity |
| October | 4213,8 | 126,415.3 | 747.7 | 0.064 | Very low scarcity |
| November | 5146.8 | 154,402.6 | 913.2 | 0.053 | Very low scarcity |
| December | 3373.3 | 101,199.1 | 598.5 | 0.080 | Very low scarcity |

**Table 5.** Environmental Sustainability for the Water Pollution Level ($WPL_{gray}$).

| Month | Average Monthly Flow Magdalena River ($m^3$) | Monthly Flow ($m^3$) | $Av_{blue}$ ($m^3$) | $WPL_{gray}$ | Sustainability |
|---|---|---|---|---|---|
| January | 1998.9 | 59,965.7 | 354.7 | 9.72 | Severe water pollution index |
| February | 2001.9 | 60,056.9 | 355.2 | 9.71 | Severe water pollution index |
| March | 3367.6 | 101,028.0 | 597.5 | 5.77 | Severe water pollution index |
| April | 4619.3 | 138,580.2 | 819.6 | 4.21 | Severe water pollution index |
| May | 4831.9 | 144,959.4 | 857.4 | 4.02 | Severe water pollution index |
| June | —— | —— | —— | —— | —— |
| July | 3940.3 | 118,208.5 | 699.1 | 4.93 | Severe water pollution index |
| August | 4004.9 | 120,145.5 | 710.6 | 4.85 | Severe water pollution index |
| September | 3672.7 | 110,181,6 | 651.7 | 5.29 | Severe water pollution index |
| October | 4213.8 | 126,415.3 | 747.7 | 4.61 | Severe water pollution index |
| November | 5146.8 | 154,402.6 | 913.2 | 3.78 | Severe water pollution index |
| December | 3373.3 | 101,199.1 | 598.5 | 5.76 | Severe water pollution index |

The situation exposed can be attributed to the fact that, since the origins of the Fátima, the Baquero wetland has been the primary wastewater recipient, leading to deterioration over time. Likewise, it can be related to the pressure on water resources exerted by other stakeholders in the area, such as African palm farmers and ranchers, especially in dry periods. Pfister et al. [78] emphasize that $SI_{blue}$ contributes not only to establishing productive systems related to water but also helps to manage water resources within a region limited by water scarcity, such as the one found in this study.

Summarizing, the $WF_{total}$ and the $SI_{blue}$ can be complementary tools that provide information that helps identify strategies to reduce the pressure over water resources in low-income communities from tropical developing countries. However, these strategies require defining guidelines that facilitate an efficient and equitable allocation of responsibilities among the agents involved, such as the water service provider, users, and environmental authorities. These stakeholders should cooperate under a water-governance scheme that minimizes anthropogenic impacts on relevant ecosystems, such as the Baquero wetland in Colombia. In addition to the above, to improve the results of this research, future work should assess the economic and technical feasibility of implementing efficient water use practices at the household level, given that these actions were identified as essential strategies to move towards sustainability in the study area.

Finally, the WF and the $SI_{blue}$ are instruments with the potential to consolidate sustainable consumption policies from the perspective of water demand. In Colombia, IDEAM has shown the relationship between water consumption and Gross Domestic Product (GDP) in different regions, identifying that the GDP increases as water consumption reduces [79]. This relationship shows the importance of changing traditional consumption models in the regions and the relevance of a regulatory framework that boosts these changes. This process of policy change is led by the Ministry of Environment and Sustainable Development, the rector of environmental affairs in Colombia. This entity formulated the National Policy for Sustainable Production and Consumption and the National Policy for Water Resources. The first policy aims to promote change in the Colombian economy's production and consumption patterns, looking for environmental sustainability and contributing to

increasing regional competitiveness. The second policy defines the objectives, strategies, goals, indicators, and strategic actions for water resources management in the country [80].

## 4. Conclusions

Conclusions from this research are:

The methodological adaptation for the estimation of the Water Footprint in the context of a low-income urban area from a tropical developing country in a data-scarce context allowed us to examine food consumption habits and intrahousehold water use practices according to the economic conditions and to establish managerial proposals that can be implemented by homeowners, and water service providers, among others. Researchers or service providers can replicate this methodological adaptation in areas lacking information, such as this case study.

The study reveals the water consumption patterns in the area according to socioeconomic level. The largest amount of water used was associated with food consumption (86.7% of the total footprint), with the highest socioeconomic stratum making the greatest contribution.

The gray water footprint was 13.2% of the total water footprint, with an annual average pollution index of 5.2. This result indicates that the Baquero wetland daily received untreated wastewater discharges from all socioeconomic levels (strata). Thus, the amount of water supplied would need to be quintupled to bring this receiving body to the conditions established by current Colombian regulations. This situation highlights the need to install a wastewater collection and treatment system for the area because, if the current conditions persist, the wetland will not be able to maintain its ecological balance in the future due to the physicochemical characteristics of the wastewater discharged by the city.

Capacities must be strengthened to motivate the adoption of water conservation practices in the community, differentiated according to the socioeconomic level (strata), such as the use of high-water consumption devices (i.e., conventional showers, taps, and toilets), increase the willingness to reuse water for domestic purposes, and in some cases, avoid using water from the water supply system to wash cars.

Adopting water management strategies would reduce household water consumption, promote a behavioral change in unsustainable water consumption patterns, and control wetland pollution. These strategies could include the implementation of water metering and water conservation campaigns by the water company as part of a program of institutional strengthening and water governance in small urban centers such as Gamarra.

**Supplementary Materials:** The following supporting information can be downloaded at: https://www.mdpi.com/article/10.3390/su15097115/s1, Figure S1: Sampling points of water quality monitoring in the critical pollution areas in El Baquero wetland; Figure S2. Number of inhabitants by socioeconomic stratum; Figure S3. Distribution of the population (adults and children) according to socioeconomic stratum; Figure S4. Distribution of the pet population according to socioeconomic stratum; Table S1. Mean ($\mu$) of the water footprint by activities (m$^3$/month); Table S2: Pearson correlation matrix for Water Footprint types; Table S3: Pearson correlation matrix for Blue Water Footprint types; Table S4: Pearson correlation matrix for the types of Green Water Footprints; Table S5: Pearson correlation matrix for the types of Gray Water Footprints.

**Author Contributions:** All authors contributed to the study's conception and design. Material preparation, data collection, and analysis were performed by B.A.P.-O., J.S.-P., A.R.-S., K.F.S.-Q., R.S.-E., I.C.D.-R. and A.S. The first draft of the manuscript was written by B.A.P.-O., J.S.-P. and R.S.-E. and all authors commented on previous versions of the manuscript. All authors have read and agreed to the published version of the manuscript.

**Funding:** This work was funded and supported by Universidad Popular del Cesar- Sectional Aguachica: Agreement No. 13 of 30 December 2021. In addition, funding and support were provided by Universidad Industrial de Santander under grant No FM-2022-01.

**Institutional Review Board Statement:** Not applicable.

**Informed Consent Statement:** Informed consent was obtained from all subjects involved in the study.

**Data Availability Statement:** All data generated or analyzed during this study are included within the article.

**Acknowledgments:** The authors thank the Universidad Popular del Cesar- Sectional Aguachica (Colombia) for funding the research seedbed project Agreement No. 13 of 30 December 2021: "Valorización y Aprovechamiento de Biomasa Residual por Bioprocesos Ambientales". The authors also thank Universidad de Investigación y Desarrollo (Colombia) for the time provided to Jonathan Soto-Paz to contribute to the development of this research. Likewise, thanks to the Gamarra water company and the inhabitants of the Fátima neighborhood for their active participation during the project. Isabel Domínguez thanks the support of the Universidad Industrial de Santander (Colombia) under grant FM-2022-01.

**Conflicts of Interest:** The authors declare no conflict of interest.

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
