# Peer review of "Assessment of the Water Footprint in Low-Income Urban Neighborhoods from Developing Countries: Case Study Fátima (Gamarra, Colombia)"

_sustainability, doi:10.3390/su15097115_

Round 1

Reviewer 1 Report

The authors estimate WF components and the sustainability level of water use in a tropical city. As such, results refer to the specific case study, while implications and the possible transferability of conclusions in other similar contexts are missing and not adequately described in the conclusion section.

Similarly, the introduction does not well explain the aim of the paper and the different paragraphs are not well (conceptually) linked one another. I suggest to enrich the introduction section highlighting the value added of the study compared to the existing knowledge (literature on both developed and developing areas).

The methodological section should be improved with more details on the survey carried out, which should be included in a separate section/subparagraph rather than being mentioned only in the description of the case study area. Also descriptive statistics of sampled households and survey responses should be briefly introduced.

Results should be discussed further, especially in relation to the possible implications of different water management options, which would allow making some hypotheses on potentially new or novel policy interventions.

In conclusion, the paper is not well structured and requires major adjustment on both the content and the form. For this reason, I encourage the authors to revise before being reconsidered for publication

Reviewer 2 Report

The sustainability analysis of the Total Water Footprint (WFtotal) and the Environmental Sustainability Index (SIblue) are holistic water management tools that allow for establishing pressures over water resources. This study assessed the WFtotal and their components (Blue, Green, and Gray) in the households of a tropical city in Colombia with less than 50,000 inhabitants and sanitation deficiencies. The activities associated with intra-household water use and higher environmental impact were identified, considering socioeconomic conditions and the water quality in the receiving water body, a wetland, through the application of surveys in a sample of households.  The results showed that 86.7 % of the WF corresponds to the WFgreen, followed by the WFgray (13.2%), and finally, the WFblue (0.2%). The high value of the WFgreen is related to food patterns in the area, which varies according to socioeconomic level. Likewise, the SIblue shows that the Baquero wetland presents scarcity scenarios regarding water quality and sedimentation, threatening the environmental service provision from this strategic ecosystem.

Overall, it is good attempt to calculate the footprint of some community in the study area. The process can serve as a guideline for other future studies. I would like to recommend the article with some modifications as under:

1. Title is too long and it may be shortened. Please reconsider the part of title as "tropical cities under 3 50,000 inhabitants". In my opinion it may be omitted.

2. Whole section 2 needs to be precise. It covers 4 pages, I recommend it be below 2 pages with very essential details, otherwise references of such studies are very common.

3. Authors should justify that they took data from many tropical regions, so far the work is showing Fatema Neighborhood.

4. Conclusions should be concise and rewritten.

Round 2

Reviewer 1 Report

The authors addressed the major concerns expressed in the first review round. The supplementary materials provided allow a better understanding of results and support scientific soundeness.

I only suggest authors to consider using findings to derive some more policy implications for both the local and similar contexts.

Reviewer 2 Report

No further comments
